# Past, Present and Future Anti-Obesity Effects of Flavin-Containing and/or Copper-Containing Amine Oxidase Inhibitors

**DOI:** 10.3390/medicines6010009

**Published:** 2019-01-15

**Authors:** Christian Carpéné, Nathalie Boulet, Alice Chaplin, Josep Mercader

**Affiliations:** 1Institute of Metabolic and Cardiovascular Diseases, INSERM, UMR1048, Team 1, 31432 Toulouse, France; nathalie.boulet@inserm.fr; 2I2MC, University of Toulouse, UMR1048, Paul Sabatier University, 31432 Toulouse Cedex 4, France; 3Cardiovascular Research Institute, School of Medicine, Case Western Reserve University, Cleveland, OH 44106, USA; amc315@case.edu; 4Department of Fundamental Biology and Health Sciences, University of the Balearic Islands, 07122 Palma, Spain; josep.mercader@uib.es; 5Balearic Islands Health Research Institute (IdISBa), 07122 Palma, Spain

**Keywords:** adipose tissue, obesity, amine oxidase, food intake, energy balance, phenelzine, semicarbazide

## Abstract

**Background:** Two classes of amine oxidases are found in mammals: those with a flavin adenine dinucleotide as a cofactor, such as monoamine oxidases (MAO) and lysine-specific demethylases (LSD), and those with copper as a cofactor, including copper-containing amine oxidases (AOC) and lysyl oxidases (LOX). All are expressed in adipose tissue, including a semicarbazide-sensitive amine oxidase/vascular adhesion protein-1 (SSAO/VAP-1) strongly present on the adipocyte surface. **Methods:** Previously, irreversible MAO inhibitors have been reported to limit food intake and/or fat extension in rodents; however, their use for the treatment of depressed patients has not revealed a clear anti-obesity action. Semicarbazide and other molecules inhibiting SSAO/VAP-1 also reduce adiposity in obese rodents. **Results:** Recently, a LOX inhibitor and a subtype-selective MAO inhibitor have been shown to limit fattening in high-fat diet-fed rats. Phenelzine, which inhibits MAO and AOC, limits adipogenesis in cultured preadipocytes and impairs lipogenesis in mature adipocytes. When tested in rats or mice, phenelzine reduces food intake and/or fat accumulation without cardiac adverse effects. Novel amine oxidase inhibitors have been recently characterized in a quest for promising anti-inflammatory or anti-cancer approaches; however, their capacity to mitigate obesity has not been studied so far. **Conclusions:** The present review of the diverse effects of amine oxidase inhibitors impairing adipocyte differentiation or limiting excessive fat accumulation indicates that further studies are needed to reveal their potential anti-obesity properties.

## 1. Introduction

The wide family of sympathomimetic agents has been a continuous source of anti-obesity drugs for decades, essentially because it encompasses at least two types of molecules that interfere with energy balance: those able to limit energy intake (exhibiting anorectic properties); and those that increase energy expenditure (exhibiting thermogenic properties). Furthermore, several of these drugs even share both activities. Numerous sympathomimetic agents have been studied so far for their anti-obesity potential (monoaminergic receptor agonists and antagonists, precursors of aminergic transmitters, neuronal reuptake inhibitors and enhancers of amine vesicular release); however, for the purpose of this review, only the inhibitors involved in the prevention of the inactivation of aminergic neuromodulators will be considered. Thus, we will focus on the compounds that inhibit amine oxidase activities and therefore contribute to impair the “turn off” of the signalling neurotransmitters norepinephrine, dopamine and serotonin. However, before reviewing this subclass of pharmacological agents that could be useful to treat obesity, it is worth mentioning that the concept of “burning fat” should be revised. Any novel anti-obesity strategy has to limit fat deposition without obliterating adipose tissue totally. On one hand, numerous mouse models with genetic ablation of white adipose tissue (WAT) have unanimously indicated that life without adipose depots is extremely unhealthy and hazardous. On the other hand, intense lipid mobilisation is more likely to lead to lipotoxicity, ectopic fat deposition and insulin resistance rather than to massive energy dissipation and safe or permanent body mass loss. Therefore, nowadays, an adequate anti-obesity treatment or prevention should limit lipid accumulation in a moderate but sustained way rather than promising an explosive fat utilisation.

Bearing this in mind, it is easier to understand how the powerful anorectic effects of amphetamines and monoamine oxidase inhibitors (MAOIs) observed in animal models for over half a century [1,2] did not result in safe therapeutic approaches for obese patients. In spite of the strong slimming efficiency of the combination of amphetamines and monoamine oxidase blockers, widely prescribed as antidepressant drugs, their mixed use was early considered as a contraindication in order to avoid overstimulation of the sympathetic system and serious cardiovascular events. At that time, drugs inhibiting neuronal monoamine uptake or evoking monoamine release were hypothesised to be more convenient strategies to limit obesity than MAO inhibitors, which were supposed to rely only on the blockade of neurotransmitter degradation. Moreover, numerous psychopharmacological studies revealed that, contrarily to substituted amphetamines, MAOIs did not reduce hunger perception in depressive patients and were not of sufficient interest to treat obesity. In fact, MAOIs have never been reported to clearly promote weight loss in depressed patients. For example, phenelzine (and, to a lesser extent, tranylcypromine) was shown to increase carbohydrate craving and to facilitate weight gain [3]. Because many psychiatric disorders are associated on their own with weight changes and dysregulated appetite control, it can be considered now that the treatment of depressed patients with MAOIs or other antidepressants is probably not the most adequate situation to validate the anti-obesity capacity of these drugs. Lastly, it is noteworthy that tricyclic antidepressants tend to cause more weight gain than MAOIs in depressed patients [3,4]. Nevertheless, various tricyclic antidepressants have been reported to act as thermogenic agents in animal models, capable of reducing body weight gain without altering food intake [5]. More recently, the anorectic drug phentermine, which is a substituted amphetamine still prescribed for weight loss, has been shown to also inhibit human monoamine oxidase (MAO) at relatively high doses [6]. A similar observation was made for fenfluramine [7], another appetite suppressant successfully used to treat obesity before being withdrawn in view of an increased risk of cardiac valvular diseases. Hence, one can deduce that, among the abovementioned sympathomimetic agents, it remains difficult to distinguish between those involved in weight loss and those inhibiting MAO.

Thus, this review presents a series of independent observations showing that an inhibition of amine oxidases can be concomitant with a reduction of fat accumulation, at least in laboratory rodents. Then, although far from recommending immediately the treatment of obese patients with amine oxidase inhibitors, we will discuss the current evidence that gives interest to novel anti-obesity strategies based on the inhibition of members of the amine oxidase family. To this aim, it is necessary to briefly describe the classification and biological functions of amine oxidases, including a presentation of their substrates and inhibitors, before reviewing the current knowledge and establishing hypotheses linking amine oxidase inhibition and energy balance control.

## 2. Flavin-Containing and Copper-Containing Amine Oxidases

Monoamine oxidases are enzymes capable of removing an amine group from a substrate (mostly a small and soluble molecule) in the presence of oxygen, and producing the corresponding ketone or aldehyde, ammonia and hydrogen peroxide. MAO proteins exist in two major forms, MAO-A and MAO-B, encoded by two genes located on the X chromosome. Both contain flavin adenine dinucleotide (FAD) as a covalently bound cofactor and share about 70% of their peptide sequence. These flavoenzymes are key regulators for normal brain function, since they contribute to the catabolism of amine neurotransmitters (essentially norepinephrine, dopamine and serotonin for MAO-A and dopamine for MAO-B, see Table 1). Accordingly, various mouse strains with genetically invalidated MAO-A exhibit increased impulsive aggressiveness, and pharmacological inhibitors of MAO modify the mood and behaviour of treated patients. However, MAOs also catalyse the oxidative deamination of many other biogenic or xenobiotic amines and they are highly expressed in peripheral tissues. Indeed, the existence of multiple forms of mitochondrial MAO was described in liver decades ago [8], while the predominance of the MAO-A form over MAO-B in human adipocytes was reported later in the 1990s [9]. Among the numerous MAO inhibitors, clorgyline can be considered as representative of the selective MAO-A inhibitors, while selegeline (also named deprenyl) is selective for MAO-B. Moreover, “older” molecules, such as pargyline, phenelzine and tranylcypromine, are nonselective MAO-A/MAO-B inhibitors [10].

Lysine-specific demethylases (LSD1 and LSD2) are homologous to MAOs in their structure and inhibition properties. These flavoenzymes are involved in gene expression regulation by removing methyl groups of histones in a reaction requiring their FAD cofactor. Since LSDs are present in chromatin and participate in epigenetic reprogramming by turning off the action of histone methyltransferases, they are currently the targets of various anti-cancer drugs under development [11]. A growing number of tranylcypromine derivatives is selected via pharmacological screening, based on their capacity to inhibit LSDs [11,12]. Another class of FAD-amine oxidase is represented by the polyamine oxidases: the constitutively expressed N1-acetylpolyamine oxidase (APAO) and the inducible spermine oxidase (SMO) [13], both being inhibited by the bis (buta-2,3-dienyl) butanediamine MDL72527 (Table 1).

However, many peripheral tissues and organs also express other amine oxidases that do not use FAD as a cofactor to perform oxidative deamination: the copper-containing amine oxidases (CAO or AOC). One of them is currently known as vascular adhesion protein-1 (VAP-1), which was initially called semicarbazide-sensitive amine oxidase (SSAO) since it was inhibited by semicarbazide but not by classical MAOIs. The SSAO/VAP-1 cupro-enzyme exhibits multiple functions. In addition to its capacity to contribute to the rolling and adhesion of circulating leucocytes to endothelial cells at the sites of inflammation [14], it is present in: (1) liver, where it is involved in inflammation and fibrosis [15] by attracting pro-inflammatory immune cells; (2) adipose tissue, where it mediates several insulin-like effects of amine substrates by producing hydrogen peroxide when degrading them at the surface of adipocytes [16]; and (3) arteries and articular cartilage, where it is involved in the maturation of extracellular matrix [17] by facilitating the cross-linking of collagen and elastin, together with lysyl oxidase (LOX, see below). SSAO /VAP1 is encoded by the third of four AOC genes found in the human genome [18], named AOC3 and located on chromosome 17. The cofactor of this amine oxidase is a post-traductionally modified tyrosine residue named topaquinone that binds copper in coordination with histidines in the catalytic domain. The two other topaquinone-containing amine oxidases are AOC1, a diamine oxidase sometimes called histaminase and AOC2, a so-called retina-specific amine oxidase. All the AOCs share common inhibitors that are often more potent than the one originally used, semicarbazide (Table 1). Consequently, SSAO has been renamed Primary amine oxidase (PrAO) [19].

The lysyl-oxidase LOX and four LOX-like proteins are also copper-containing amine oxidases that oxidize lysine and hydroxylysine residues of insoluble substrates, such as collagens and elastin, and other soluble substrates as well [20]. As mentioned above, LOX collaborates with AOCs in the maturation of extracellular matrix. It must be pointed out that LOX was also known as Ras recision gene, a tumour suppressor, the expression of which is deeply altered in tumour tissues. Briefly, LOX and LOXLs can be considered as metastatic promoters: they increase in invasive and metastatic cancers, triggering excessive crosslinking of collagen fibres and developing fibrosis.

The use of LOX inhibitors derived from β-aminopropionitrile (β-APN) is recognised to limit metastasis, and therefore could restrict obesity-associated cancers, such as those affecting colon or liver, among others.

A list of amine oxidase inhibitors for the different forms of MAOs, AOCs, APAO, SMO and LOX–LOXLs is included in Table 1, albeit not an exhaustive one. In fact, numerous inhibitors are not specific to only one type of amine oxidase, while several are irreversible and others reversible in their inhibitory action. The chemical structures of several amine oxidase inhibitors are given in Table 2. Moreover, their mode of inhibition could be related to the binding to the catalytic site or allosteric in nature, or even combined. To simplify, semicarbazide and many other carbonyl reagents are able to block AOCs without altering MAO and FAD-containing amine oxidases, while pargyline, clorgyline and selegiline only block MAO. Moreover, it must be noted that phenelzine is an inhibitor of both MAOs and AOCs, as is the case for tranylcypromine. Additionally, the pharmacological properties of these inhibitors may present substantial interspecies differences, as is the case for the substrates indicated in Table 1. For clarity, it is necessary to mention here that tyramine is a substrate of both murine MAO and SSAO, while it is mainly oxidized by MAO in humans. Similarly, benzylamine is an SSAO substrate of reference that might be oxidized by MAO-B too.

## 3. Past and Present Observations of Amine Oxidase Inhibitor Effects on Adiposity in Animal Models

A screening of thermogenic drugs as potential anti-obesity agents performed by Dulloo and Miller revealed that, among more than 30 sympathetic stimulants, MAOIs were endowed with a remarkable capacity to increase energy expenditure and reduce fat without loss of body protein in various mouse and rat models of obesity [2]. In particular, it was found that the drug tranylcypromine reduces fat content in rats and mice with either genetic or diet-induced obesity to the same extent as ephedrine. Since tranylcypromine, also known as 2-phenylcyclopropylamine, is a nonselective and irreversible MAOI originally thought to be an amphetamine analogue, the authors concluded that its notable anti-obesity effect was due to its capacity to increase neurotransmitter half-life and energy expenditure. Moreover, when given in rodent chow at 500 mg/kg diet for 7–9 weeks, tranylcypromine is as highly thermogenic as ephedrine at 1000 mg/kg diet [2]. Nevertheless, these results could be interpreted in a different light, since: 1) ephedrine is currently banned in view of its amphetamine-like structure and related adverse effects; and 2) tranylcypromine is not a simple MAOI, being now recognized as an SSAO [33] and LSD inhibitor [11,12]. Furthermore, recent studies have confirmed an anti-obesity effect for tranylcypromine even in mice invalidated for the uncoupling protein-1 (UCP1) involved in non-shivering thermogenesis and in beiging of WAT [26]. Thus, these recent findings indicate that tranylcypromine not only increases the basal metabolic rate, but it also increases spontaneous locomotor activity and decreases food intake. Therefore, among the studies of tranylcypromine derivatives, currently boosted by the promising inhibition of tumour progression and metastasis that generates the inactivation of LSD1, the observation of putative anti-obesity properties deserves further attention.

The MAO-B inhibitor selegiline was very recently reported by Nagy and co-workers to reduce adiposity induced by a high-fat, high-sucrose diet in male rats without influencing body weight [25]. Similarly, the MAO inhibitor phenelzine has been shown to reduce body fat content in mice fed a standard rodent chow without a decrease in body weight gain or food intake [22]. These recent independent observations are in line with our findings indicating that pargyline, another MAOI, is able to reduce body weight gain in female obese Zucker rats when administered by a daily intraperitoneal injection (Figure 1). Treatment with pargyline also inhibited the tyramine oxidation by MAO in adipose tissue [21]. However, since SSAO/VAP-1 is another amine oxidase highly expressed in adipocytes, and since its substrates can mimic several insulin effects, such as stimulation of glucose transport [34], we also associated the SSAO inhibitor semicarbazide to the treatment performed with pargyline and observed a potentiation of their anti-obesity effects (Figure 1). Such combined treatment limited the antilipolytic effect of a high dose of tyramine (1 mM) when tested in vitro on isolated adipocytes [21].

However, these observations are not the first indicating that an inhibition of amine oxidases other than MAO could be effective in limiting fattening in treated animals. To the best of our knowledge, the first observations indicating that repeated administration of an SSAO inhibitor limits weight gain were serendipitously made by Yu and co-workers in 2004 [31]. This group was investigating the involvement of AOC-mediated deamination on cardiovascular events and observed that treatment with the SSAO inhibitor (E)-2-(4-fluorophenethyl)-3-fluoroallylamine (FPFA) reduces weight gain in obese and diabetic KKAy mice.

Interestingly, other drugs initially studied for their beneficial actions on vascular complications have been shown to produce SSAO inhibition and alter fat content in treated animals. Besides its multiple cardiovascular effects, hydralazine was shown to inhibit SSAO [35,36], as well as to inhibit the benzylamine antilipolytic effect in adipocytes [16]. With the aim to determine whether hydralazine is a beneficial antihypertensive therapy during obesity development, Carroll and co-workers found serendipitously that hydralazine treatment lowered body fat content in obese rabbits [30]. Similarly, aminoguanidine, a multipotent drug able to block nitric oxide synthases, has been shown to irreversibly inhibit copper-containing amine oxidases, including SSAO [37]. Thereafter, it was observed that prolonged aminoguanidine treatment slightly reduces fat deposition in obese Zucker rats [29]. Again, there was a strong inhibition of adipocyte SSAO/VAP-1 activity and an impairment of insulin-like effects of SSAO substrates in fat cells from aminoguanidine-treated rats [29].

Since all these molecules sharing SSAO inhibitory properties exhibited some anti-obesity effects, it was of utmost interest to check whether semicarbazide, the reference agent for inhibiting copper-containing oxidases, was unambiguously reported to reduce body weight and/or adiposity. Indeed, it was independently observed that semicarbazide reduces body weight gain in Sprague Dawley and in Brown Norway rats in response to repeated intraperitoneal administration at 900 µmol/kg body weight/day for 6–8 weeks [17,24,38]. We also observed that, when added at 0.125% to the drinking water of mice between the 5th to the 13th week of age, semicarbazide limited food intake, body weight gain and adiposity [28]. Moreover, Takahashi and co-workers showed that semicarbazide dramatically reduced body weight gain in rats, when added to their food at 0.1% [27]. A semicarbazide “slimming” effect was detected from the first week of treatment until the end of experiment 12 weeks later. However, such prolonged treatment, corresponding to an ingested dose estimated between 700 and 1000 µmol/kg bw/d, revealed toxicological effects of semicarbazide, including deformation of connective tissues and articular cartilage, together with a loss of bone mass. Of note, a lower dose of semicarbazide did not provoke such adverse effects but was devoid of any slimming action, as reported in Figure 1, for semicarbazide treatment at 100 µmol/kg/d in genetically obese rats. Nevertheless, such treatment completely inhibited SSAO activity in WAT [21] and potentiated the inhibition of weight gain induced by pargyline (Figure 1). From these observations and a survey of the literature, it could be concluded that semicarbazide chronic treatment reduces fat deposition but is rather toxic. Moreover, SSAO inhibition is necessary but not sufficient for the unanimously observed anti-obesity properties of high semicarbazide dosages.

## 4. Reducing the Risk/Benefit Ratio of Semicarbazide by Combining with Other Inhibitors

Semicarbazide is a small urea derivative (NH_2_-C=O-NH-NH_2_) belonging to the chemical family of hydrazines, and is thus a carbonyl reagent that binds to the SSAO active site, but which can react with many other targets. As a consequence of its use in industrial organic chemistry, it is now banned as food contaminant (suspected genotoxic) and considered as a marine pollutant capable of disturbing endocrine and reproductive systems in fishes [39,40]. When administered at high doses, it limits growth and alters joints and cartilage in rodents [17] while other hydrazine derivatives (hydrazine, acetylhydrazine and 1,2-dimethylhydrazine) are colon-specific carcinogens. Intriguingly, studies of hepatic gene expression and proteomics have revealed that, among the major toxic effects of hydrazine in liver, there is altered expression of proteins related to lipid transport and metabolism [41,42]. To circumvent these difficulties, it was necessary to reduce semicarbazide administration below its No Observable Adverse Effect Level (NOAEL) established at 250 ppm [27] by potentiating its actions, or to replace it with less-toxic hydrazine derivatives and/or other AOC inhibitors.

As illustrated in Figure 1, a combination of a low dose of semicarbazide and pargyline resulted in a potentiation of their anti-obesity effects in obese Zucker rats. Indeed, the reduction of fat deposition by intraperitoneal semicarbazide 100 µmol/kg bw/d + pargyline 60 µmol/kg bw/d was not due solely to food intake inhibition, since, at the end of treatments, obese rats treated by combined administration of AO inhibitors presented less fat than the pair-fed control [21]. In Wistar rats, chronic treatment with a low dose of semicarbazide and pargyline also induced a significant reduction of body weight gain and adiposity [24]. Our proposed rationale for this potentiation is that both MAO and SSAO converge in producing hydrogen peroxide when oxidizing their substrates; this in turn generates more oxidation product to mimic several of the insulin actions in rat adipocytes [43]. In keeping with this, it was necessary to block both MAO and SSAO to limit the antilipolytic and lipogenic effects of tyramine in rodent adipocytes [24]. Thus, the link between amine oxidase inhibition and body mass control was not solely limited to actions on neuronal transmission but also appeared to directly influence adipocyte metabolism.

Curiously, the LOX inhibitor β-aminopropionitrile also enhanced the limitation of fat accumulation by semicarbazide [24,44]. More recently, LOX inhibition has been reported to reduce body weight gain and to improve the metabolic profile in diet-induced obesity in rats when orally administered with β-aminopropionitrile at 100 mg/kg bw/d [32]. In this study, β-aminopropionitrile also attenuated the increase in WAT fibrosis. Indeed, β-aminopropionitrile inhibits a LOX activity that increases with hypoxia as a result of the upregulation of hypoxia-inducible factor (HIF-1) during WAT hypertrophy, i.e., when the adipocyte size enlarges [45]. Unfortunately, β-aminopropionitrile is also toxic for joints and bones, and alters aortic stiffness [38].

Not all copper-containing amine oxidases inhibitors have been reported to decrease body weight gain. On the contrary, the SSAO/VAP-1 inhibitor LJP 1207 prevented the acute weight loss observed in mice with oxazolone-induced colitis [46]. However, in these experiments, the challenge was not obesogenic at all: the weight loss exhibited by control group was caused by an extreme inflammatory disease and a catabolic condition, while the anti-inflammatory action of SSAO/VAP-1 blockade resulted in a significant recovery. Anyhow, and as proposed above for novel LSD inhibitors, the putative anti-obesity properties of AOC inhibitors warrant further studies.

## 5. Combining the Inhibition Effect of MAOs and AOCs on Adiposity: The Case of Phenelzine

As indicated above, the non-selective MAO inhibitor phenelzine is also an SSAO inhibitor [10]. In addition, owing to its hydrazine structure and not to MAOI considerations, phenelzine has been reported to induce hypoglycaemia in animal models and in depressed patients through the impairment of gluconeogenesis, likely as a consequence of phosphoenolpyruvatecarboxykinase (PEPCK) inhibition [47]. Since phenelzine is an old drug prescribed for antidepressant therapy, with no clear evidence of the serious adverse effects attributed to hydrazines as discussed above, it is a good candidate for drug repositioning. We have reported that phenelzine treatment reduces visceral WAT in obese rats [24] and subcutaneous WAT in non-obese mice [22], reinforcing the idea that combined inhibition of MAO and AOCs was instrumental for impairing fat deposition. Moreover, phenelzine dose-dependently prevents in vitro adipogenesis in various models of cultured preadipocytes by repressing the sterol regulatory element-binding protein-1c, which drives lipogenic pathways [48]. The direct inhibition of insulin-stimulated lipogenesis by phenelzine in mature adipocytes further supported an “adipocentric hypothesis” for its limitation of fat deposition [49]. Unfortunately, our attempt to improve such anti-lipogenic/adipogenic effects in mice by a combined treatment with resveratrol, a natural molecule known to share similar actions, was unsuccessful [50]. Nevertheless, we recently observed that phenelzine not only hampers PEPCK activity in adipocytes, but also downregulates its expression [51]. Further studies are warranted to better understand the interests and limits of this agent and other inhibitors of amine oxidases and to decipher their respective roles.

## 6. Adipocytes as Targets of Amine-Oxidase-Interacting Agents

Since fat cells are apparently the more impacted cell type in obesity, simplified models are extensively used for the screening and development of anti-obesity drugs. Consequently, cultured preadipocyte cell lines, such as 3T3-L1, are useful to determine whether a given factor is promoting or impairing lipid deposition. In spite of the limits of these models, (e.g., murine 3T3L1 cells express a panel of receptors different from that of human fat cells, and their fat droplets are multilocular while mature adipocytes are unilocular), many factors exerting an inhibitory effect on lipid accumulation in differentiating preadipocytes have been qualified as promising anti-obesity agents deserving to be further tested in vivo. Conversely, several anti-obesity drugs that were developed in view of their in vivo capacity to reduce appetite, such as sibutramine or rimonabant, were reported later on to be effective in such in vitro models since they also directly activate lipolysis or inhibit lipogenesis in adipocytes (for review, see a chapter by Carpéné and Iffiu-Soltész [44]). It was therefore justified to study the direct effects of amine oxidase inhibitors on fat cells. Interestingly, when differentiating into an adipocyte, the preadipocyte deeply alters the expression of copper-containing and flavin-containing amine oxidases. Even in humans, SSAO expression emerges during adipogenesis [52,53,54]. In parallel, there is a smaller increase of MAO activity in differentiating adipocytes [55]. Unfortunately, the endogenous substrates of the adipocyte SSAO/VAP-1 are far from being well-defined, and it is also the case for MAO. Most importantly, the addition of exogenous SSAO substrates to the culture media (e.g., benzylamine) can partially replace or reinforce the insulin stimulation of adipogenesis [53,56]. It is therefore not surprising to note that most of the tested amine oxidase inhibitors have been reported to directly interact with adipocyte metabolism. Hence, phenelzine inhibits adipogenesis in various models of cultured preadipocytes, as is the case for pargyline and semicarbazide [48]. Tranylcypromine inhibits LSD1 and reduces adipogenesis in cultured 3T3-L1 adipocytes, together with increased expression of energy expenditure [57]. However, β-aminopropionitrile improves insulin-stimulated glucose uptake when this cell line is subjected to TNFα-induced insulin resistance [32]. Similarly, the selective MAO-A inhibitors moclobemide and Ro41-1049, and the MAO-B inhibitor selegiline, promote adiponectin production during adipocyte differentiation in human adipose mesenchymal stem cells [23].

Lastly, a very recent report by Yang and co-workers indicates a decreased glucose utilisation and increased lipid trafficking in preadipocytes when adipogenesis-dependent SSAO emergence is defective, therefore confirming that SSAO is a regulator of energy utilisation processes in adipocytes [58]. Moreover, observations from the genetic total invalidation of AOC3 in mice seem to support this point of view since—irrespective of the gender, strain or mode of invalidation considered—the lack of functional AOC3 is concomitant with a slight tendency of being overweight [59,60]. Such overweight is in contrast with the above-reported limitations of fat deposition found after pharmacological inhibition of SSAO and needs further clarification. Hopefully, there is a clear concordance between the increase of energy expenditure induced by chronic administration of pharmacological MAO inhibitors and the reduced body weight observed in MAO knockout mice [61].

## 7. Future Directions and Combining in Vitro, in Vivo and Clinical Studies

Is has been recognized that it is more convenient to reduce body weight gain by promoting lipid mobilisation and utilisation during the onset of obesity rather than treating massively hypertrophied WAT. Hence, the most important readouts in future clinical studies have to be the determination of the percentage of body fat and the anatomical location of the fat stores rather than the body weight itself because any efficient anti-obesity agent should be able to prevent/reduce excessive lipid storage in the visceral fat without impairing lean mass homeostasis. Obviously, a limitation of ectopic fat deposition and a reduction of the visceral/subcutaneous adipose tissue mass ratio should be demonstrated before affirming that an inhibition of amine oxidase(s) may be protective against obesity and its complications.

In humans, the development of an anti-obesity treatment based on the use of hydrazine derivatives will be probably limited by their potential toxicity, except for the abovementioned case of phenelzine. However, studies carried out in depressed patients also show that another MAOI, tranylcypromine, is not associated with weight gain or with a hepatotoxicity risk since this inhibitor is not a hydrazine derivative [3,4]. As tranylcypromine is an SSAO and LSD inhibitor, there is therefore an opportunity to check the potential anti-obesity properties of this drug and its novel derivatives in studies performed to test anti-cancer or anti-inflammatory properties. In keeping with this, there are various other drugs or compounds known to mitigate excessive weight gain, which have been demonstrated to inhibit MAO or AOC activity. Among these, caffeine may be of interest, being the most widely consumed natural alkaloid, since it exhibits a more or less elusive anti-obesity action in consumers and inhibits SSAO [19]. This is also the case for other methylxanthines, including theobromine, recently reported to inhibit bovine SSAO [62], whereas a methylxanthine thermogenic action has been attributed so far to phosphodiesterase inhibition or to adenosine receptor antagonism, both resulting in higher intracellular cAMP levels [63]. Lastly, resveratrol and other natural polyphenols, able to mitigate obesity in many animal models [64], have been reported to inhibit MAO [65,66]. Whether amine oxidase inhibition contributes to the role of these agents in the control of body mass or in their anti-adipogenic effects [67,68] remains to be determined.

On the other hand, most of the MAOIs or SSAO/VAP-1 inhibitors listed in this review also interact with other multiple targets. It cannot be ruled out that additional mechanisms other than the blockade of oxidative deamination are involved in their observed promising anti-obesity effects (e.g., phenelzine also interacts with the GABAergic system). It should be of utmost importance to determine whether there is a substantial spontaneous activity of these amine oxidases in obese individuals, since in the absence of a pathologically elevated activity, no inhibitor can induce substantial change in the function of its target(s) and downstream events. In fact, elevated SSAO expression/activity has been reported in WAT animal models of genetic [69] or diet-induced obesity [70]. In humans, there is a soluble form of SSAO circulating in plasma, which does not seem to vary with obesity [71], but which is largely less abundant than in adipose tissue. Nevertheless, an increased MAO activity has been reported in adipocytes from obese individuals [72], leaving open the opportunity to expect a mitigation of obesity if impaired by inhibitors.

Finally, it is worth mentioning that obesity is accompanied by low-grade inflammation, and is characterized by an increased number of immune cells around the adipocytes. SSAO/VAP-1 inhibitors contribute to reduce inflammation in many experimental models [14,46] and might be useful in modulating a purported role of adipose tissue macrophages in the synthesis and degradation of lipolytic catecholamines [73,74,75]. This further supports the anti-obesity potential of MAOIs evidenced in animal models but hardly observed during clinical trials.

## 8. Conclusions

In conclusion, there are converging observations suggesting that MAOIs, known to provide clinical benefits in the treatment of mental diseases, also interact with metabolism regulation in cultured adipocytes and animal models of obesity. In addition, small molecules that inhibit SSAO/VAP-1, and that have recently been reported to be useful for the treatment of many inflammatory diseases, are also capable of influencing adipocytes, which highly express this cupro-enzyme. Although there is not yet a confirmed clinical demonstration of the usefulness of amine oxidase inhibitors for treating obesity and related complications, testing of such inhibitors (either selective or nonselective, or even used in combination) could bring about promising novel pharmacological treatments for obese subjects.

## Figures and Tables

**Figure 1 medicines-06-00009-f001:**
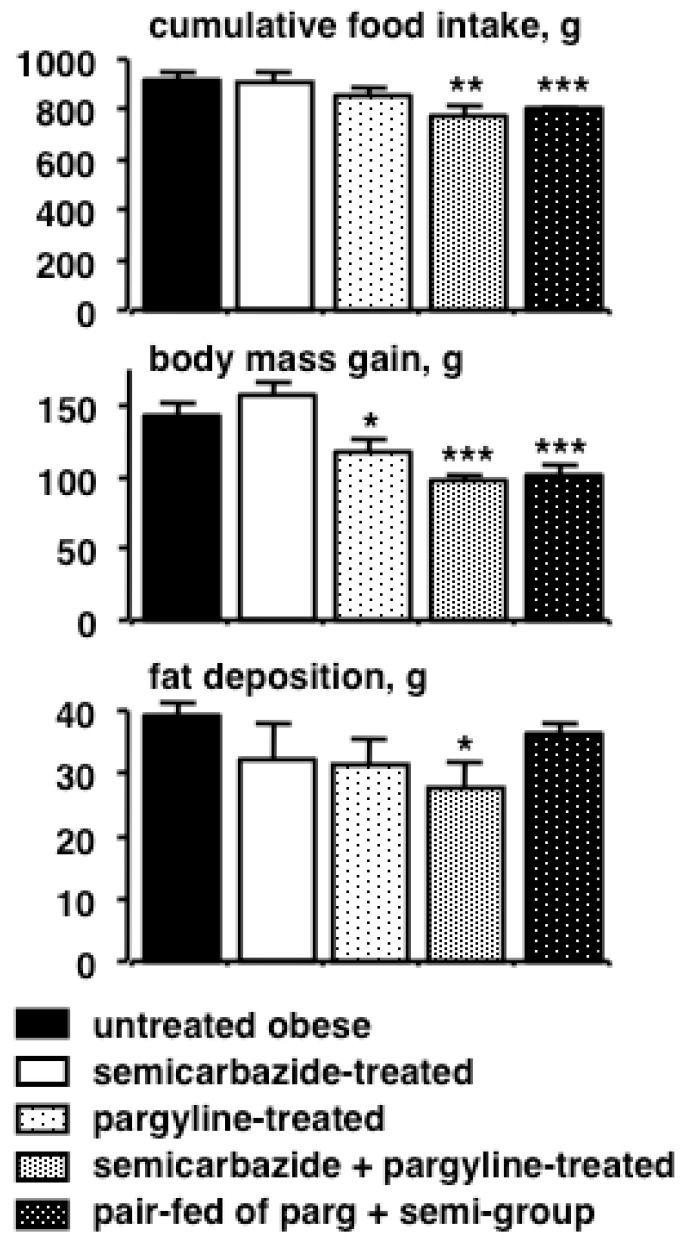
The anti-obesity effect of the combination of monamine oxidase and copper-containing amine oxidase inhibitors on weight gain and adiposity in obese Zucker rats. Nine-week old female obese rats were daily injected for 5 weeks with saline (untreated, dark column), semicarbazide (100 µmol/kg body weight/day, open columns), pargyline (60 µmol/kg/d, shaded columns) or their combination (semicarbazide + pargyline). A group of rats received the daily amount of chow consumed by the combination-treated group (pair-fed). At the end of treatment, the sum of inguinal + intra-abdominal adipose tissues removed from each rat was subtracted by 34.6 g, the equivalent mass found in the initial control group. Unpublished data given as mean ± standard error of the mean (S.E.M.) of five rats per group, for cumulative food intake, (upper graph), body mass gain (middle graph) and fat deposition (lower graph). Different from untreated at: * *p* <0.05, ** *p* <0.02, *** *p* <0.01.

**Table 1 medicines-06-00009-t001:** Mammalian amine oxidases, their substrates and the effects of their inhibitors on body weight control or in vitro adipogenesis.

Amine Oxidase	Substrate(s)	Inhibitor(s)	Effects on BW or Fat Accumulation	References
MAO-A	norepinephrine (=noradrenaline) dopamine serotonin tyramine tryptamine	pargyline phenelzine tranylcypromine moclobemide clorgyline toloxatone	↓ food intake ↓ SCWAT in mice ↓ BW gain mice ↑ adipogenesis	[21] [22] [2] [23]
MAO-B	dopamine benzylamine	pargyline tranylcypromine phenelzine deprenyl (=selegiline)	↑ effect of SCZ in reducing WAT in rats ↓ adipogenesis ↓ WAT rats	[24] [23] [25]
LSD-1	histones	tranylcypromine	↓ BW gain mice	[26]
LSD-2	histones	tranylcypromine	↓ adipogenesis	[23]
APAO	N-acetylspermine	MDL 72527		
SMO	spermine	MDL 72527		
AOC1	histamine putrescine cadaverine	semicarbazide *(others: see AOC3)*	↓ BW	[17]
AOC2	tyramine tryptamine	Semicarbazide *(others: see AOC3)*	↓ BW	[27]
AOC3	benzylamine methylamine aminoacetone	Semicarbazide phenelzine aminoguanidine tranylcypromine phenylhydrazine hydralazine FPFA BTT2052 (=SZE5302)	↓ BW gain mice ↓ SCWAT rats ↓ BW rabbits ↓ WAT mice	[28] [21] [29] [30] [31]
LOX	collagen elastin cadaverine lysine	β-aminopropionitrile semicarbazide tetrathiomolybdate	↓ BW gain DIO rats	[32]
LOXL 1-4	elastin benzylamine	β-aminopropionitrile		

MAO-A and B: Monoamine oxidases A and B; LSD-1 and -2: Lysine specific demethylases 1 and 2, APAO: N^1^-acetylpolyamine oxidase; SMO: Spermine oxidase; AOC1 and 2 and 3: Copper-containing amine oxidases 1 to 3; LOX: Lysyl oxidase, LOXL1–4: Lysyl oxidase-like 1 to 4; BW: body weight; DIO: diet-induced obesity; WAT: white adipose tissue; SCZ: semicarbazide; FPFA: (E)-2-(4-fluorophenethyl)-3-fluoroallylamine; BTT 2052: 2-(1-methylhydrazino)-1-indanol; MDL 72527: N^1^,N^4^-bis(buta-2,3-dienyl)butanediamine; (=) denotes another name of the same molecule.

**Table 2 medicines-06-00009-t002:** Amine oxidase inhibitors, names and chemical structures.

**Pargyline**	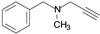	irreversible MAO-A inhibitor
**Clorgyline**	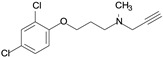	irreversible inhibitor of MAO-A
**Moclobemide**	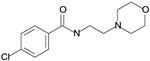	reversible inhibitor of MAO-A
***R*-deprenyl**	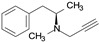	irreversible inhibitor of MAO-B
**Tranylcypromine**	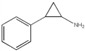	irreversible MAO inhibitor and AOC inhibitor
**Phenelzine**	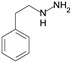	irreversible MAO inhibitor and AOC inhibitor
**Semicarbazide**	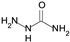	irreversible inhibitor of AOC 1–3inhibits LOX
**Aminoguanidine**	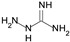	irreversible of AOC 1–3inhibits LOX
**Beta-aminopropionitrile**	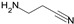	inhibitor of LOX and LOXL

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
