# Peer review of "Past, Present and Future Anti-Obesity Effects of Flavin-Containing and/or Copper-Containing Amine Oxidase Inhibitors"

_medicines, 2019, doi:10.3390/medicines6010009_

Round 1
Reviewer 1 Report
This is very good paper, well presented and described. In my opinion, it is suitable for publication in its current form after completing with Figures presenting general structures of inhibitors mentioned in the manuscript.
Author Response
This is very good paper, well presented and described. In my opinion, it is suitable for publication in its current form after completing with Figures presenting general structures of inhibitors mentioned in the manuscript.
Thank you for this positive reviewing. As proposed, a Table 2 has been added to show the chemical structure of most of the of amine oxidase inhibitors tested for their influence on energy balance.
Reviewer 2 Report
In this review article, entitled “Amine oxidase inhibitors as anti-obesity agents”, the authors introduced diverse effects of amine oxidase inhibitors on the impairment of adipocyte differentiation or the limitation of excessive fat accumulation. This manuscript has depicted interesting and useful information about the anti-obesity potential of different amine oxidase inhibitors, especially phenelzine and semicarbazide. The significant is good and should be useful for the development and study of anti-obesity agents. English and writing are impeccable. I recommend in minor revision.
1) Please define abbreviations at the first time when they appear in the abstract as well as the text in the manuscript. For example, FAD in line 15, SSAO/VAP-1 in line 21.
2) It is suggested to shorten the abstract. However, it is left up to the authors’ decision.
3) The description of semicarbazide and other molecules (lines 20 to line 28) might be revised and consolidated to main body.
4) Please consider providing full words of amine oxidases to help readers interpret the information as shown in Table 1.
Author Response
In this review article, entitled “Amine oxidase inhibitors as anti-obesity agents”, the authors introduced diverse effects of amine oxidase inhibitors on the impairment of adipocyte differentiation or the limitation of excessive fat accumulation. This manuscript has depicted interesting and useful information about the anti-obesity potential of different amine oxidase inhibitors, especially phenelzine and semicarbazide. The significant is good and should be useful for the development and study of anti-obesity agents. English and writing are impeccable. I recommend in minor revision.
Thanks for your comments. We have tried to further improve clarity and English , and all our corrections are marked in red in the revised version.
1) Please define abbreviations at the first time when they appear in the abstract as well as the text in the manuscript. For example, FAD in line 15, SSAO/VAP-1 in line 21.
We agree with referee's remark. FAD has been replaced by " flavin adenine dinucleotide" since the abbreviation has not been re-used in the abstract. "Semicarbazide-sensitive amine oxidase/vascular adhesion protein-1 (SSAO/VAP-1) " is now fully introduced at its first appearance. However, these corrections/additions highlighted in red for the sake of clarity did not help in reducing the word count of the abstract (see below).
2) It is suggested to shorten the abstract. However, it is left up to the authors’ decision.
The revised version of the abstract is now 220-word long and should fulfil therefore the required concision for retrieving in databases.
3) The description of semicarbazide and other molecules (lines 20 to line 28) might be revised and consolidated to main body.
For appearing attractive for the potential readers, we would have preferred to mention most of the inhibitor molecules, but they were moved to the main text to shorten the summary as kindly recommended by the reviewer. Thus, the following inhibitors have been removed from the abstract: fluorophenetyl-fluoroallylamine, hydralazine and aminoguanidine, as well as β-aminopropionitrile, selegiline, tranylcypromine, and pargyline.
4) Please consider providing full words of amine oxidases to help readers interpret the information as shown in Table 1..
Thanks for the input. The complete names of amine oxidases have been introduced in Table 1 legend.